# Seismic Performance Evaluation and Analysis of Vertical Hydrogen Storage Vessels Based on Shaking Table Testing

**Sangmoon Lee** [1] , **Bubgyu Jeon** [2] **and Wooyoung Jung** [3,*]

1 Institute for Smart Infrastructure, Gangneung-Wonju National University, Gangneung-si 25457, Republic of Korea; idealmoon@gwnu.ac.kr
2 Seismic Research and Test Center, Pusan National University, Busan 43241, Republic of Korea; bkjeon79@pusan.ac.kr
3 Department of Civil and Environment Engineering, Gangneung-Wonju National University, Gangneung-si 25457, Republic of Korea
* Correspondence: woojung@gwnu.ac.kr

**Abstract:** In this study, the structural integrity of a system installed on protrusion concrete, considering the usability of a vertical hydrogen storage vessel, was verified. To achieve this, a site survey was conducted to select the target structure, and analytical validation was performed to design specimens for shaking table tests. Subsequently, dynamic behavior characteristics were analyzed using an artificial earthquake simulated according to the procedures outlined in ICC-ES AC 156, which is the seismic design criterion. As a result, it was observed that the seismic motion was amplified by approximately 10 times compared to the original load magnitude, based on the acceleration response of the test specimen. It is inferred that the seismic motion occurring during an earthquake could cause significant damage to both the internal and external aspects of the structure, depending on the structure's form and the composition of materials. Through analytical verification and testing, it was revealed that the main structure of the test specimen and the anchor bolts for installation met the seismic performance criteria. However, the protrusion concrete area exhibited damage, indicating a structural vulnerability when subjected to external forces such as earthquakes. Consequently, on-site measures to address this structural risk need to be explored.

**Keywords:** hydrogen storage vessel; on-site investigation; seismic performance; shaking table test; time history analysis

## 1. Introduction

Hydrogen, which is receiving increasing attention as an alternative energy source, is an environmentally friendly element when used as a direct combustion fuel, as it only generates water and heat, without emitting pollutants into the atmosphere [1]. Hydrogen has the advantage of being easily stored in various forms, such as high-pressure gas, liquid hydrogen, and hydrogen storage alloys, allowing it to be used for a wide range of applications, from power generation to transportation and future industrial processes [2]. In particular, the outlook for the global hydrogen storage system market is forecasted to have an annual market value impact of USD 2.5 trillion by 2050. This outlook underscores the growing interest and importance of research into hydrogen energy storage systems [3]. The methods for storing hydrogen can be broadly categorized into three main approaches: liquid hydrogen storage, high-pressure gaseous hydrogen storage, and hydrogen adsorption storage, which involves chemically or physically adsorbing hydrogen onto a solid material [4]. Indeed, hydrogen, in its gaseous state, has the characteristic of a low density and large volume, making it challenging to store a significant amount of hydrogen in a limited space. For this purpose, high-pressure gaseous hydrogen is compressed and stored in vessels, requiring storage vessels capable of withstanding high pressures [5]. The shapes of these vessels are classified based on their purpose and storage capacity. The type of

storage vessel commonly used in infrastructure for storing large quantities of hydrogen is widely utilized due to its economic feasibility, ease of fabrication, and convenient installation. This type typically features a body made of easily producible and installable steel. However, due to the inherent heavy self-weight of the structure and exposure to corrosion and repetitive loading in the anchorage system, the structural integrity of vessels is relatively vulnerable compared to other vessel types [6].

Vertical vessels made of metal materials are typically supported by independent systems that connect to anchors, penetrating through the slab constructed from non-reinforced concrete and reinforced concrete. The specific configuration and height of the hydrogen storage vessel's bottom piping system are taken into account when establishing this anchorage system. This method is susceptible to various damages due to external factors such as earthquakes, depending on factors such as the self-weight of the facility element, height, and anchoring method. Among these, most damages are related to the concentration of stress in the settling components due to external forces, leading to issues such as the destruction of protrusion concrete and anchor damage [7]. Figure 1 illustrates a case of damage caused by the overturning of hydrogen storage vessels due to an earthquake.

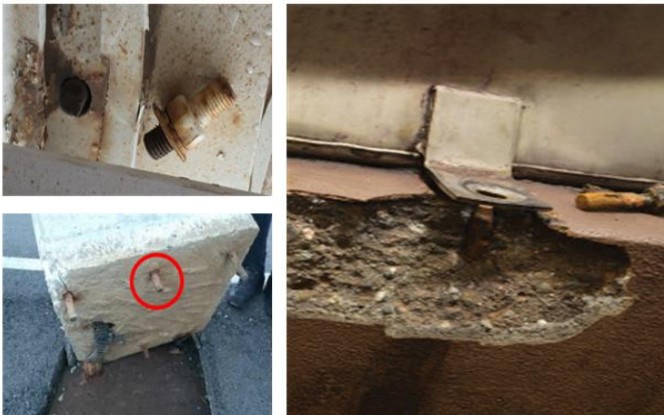

**Figure 1.** Damage to protrusion concrete caused by overturning under earthquake conditions.

The need for rigorous structural performance assessments of the target structure arises due to the potential for critical incidents such as loss of life and property damage resulting from hydrogen storage vessel damage, as mentioned in the preceding paragraph. In countries prone to frequent seismic activity, a strengthening of regulations is underway: it requires the assessment of seismic performance for critical infrastructure and equipment, with a particular focus on public buildings [8,9]. The seismic performance verification of such facilities aims not only to assess the structural safety of the facility itself but also to ultimately verify the continued functionality of its inherent features following a seismic event. In the Republic of Korea, the revised Building Seismic Design Standards (as of March 2019) stipulate that seismic performance verification should be conducted for major non-structural elements that must remain functional after an earthquake event. For critical societal structures such as power plants, rigorous seismic performance verification is required for key facilities, to ensure safety. In the case of broadcasting and telecommunication facilities, the seismic testing method for broadcasting and telecommunication equipment is typically verified through shaking table testing or analytical methods to establish the fundamental seismic design criteria. To investigate the behavior and vulnerability of structural and non-structural elements during actual earthquakes, various previous studies have employed seismic simulation methods such as shaking table testing, as well as other earthquake simulation techniques [10–13].

In this study, the goal was to conduct an analysis of the behavioral characteristics and assess the seismic performance of vertical hydrogen storage vessels during an earthquake. The selection of the target structure was based on actual on-site investigations. Subsequently, structural design was performed through numerical analysis, culminating in a

final stage of shaking table testing. The shaking table testing method followed ICC-ES AC 156 [14], and it was executed by creating a Required Response Spectrum (RRS) and artificial seismic waveforms that satisfied the common requirements of both the Building Seismic Design Standards and Seismic Design Criteria. The acceleration scaling of input ground motions was adjusted until structural damage occurred, at which point the test was concluded. Additionally, in the process of fabricating full-scale specimens for shaking table testing, a preliminary interpretative study was conducted based on site data investigations and domestic seismic design standards. The results of this numerical study, including analyses of anchor load response, seismic response characteristics, and structural acceleration response, were utilized as a basis for the fabrication process. The objective of this evaluation is to provide predictable load information transmitted to the structure for the seismic performance evaluation of various forms of hydrogen storage vessels using shaking table testing in the future.

## 2. Literature Review

In countries prone to frequent seismic activity, there is a growing trend of strengthening regulations that focus particularly on verifying the seismic performance of critical facilities and equipment, with an emphasis on social capital infrastructure. The goal of seismic performance validation for such facilities extends beyond ensuring structural integrity: it also aims to confirm the sustained functionality of systems in the aftermath of an earthquake. While the former can be verified through testing and analytical methods, the latter is commonly known to be efficiently validated through testing procedures [15–17]. In March 2019, the Republic of Korea revised its Architectural Seismic Design Standards (KDS 41 17 00) [18], specifying the requirement to substantiate the seismic performance of key non-structural elements for which functionality must be maintained post-earthquake. Nuclear power plants require rigorous verification of seismic performance for vital safety-related equipment. In the case of broadcasting and telecommunication facilities, the seismic testing methods prescribe vibration testing or analysis to validate the fundamental seismic design [19]. These regulatory measures underscore the commitment to ensuring the resilience of critical infrastructure in the face of seismic events, employing a combination of testing and analytical approaches tailored to the specific needs of different types of structures and facilities. The recently proposed ICC-ES AC 156 testing method has been chosen as one of the specialized seismic performance verification methods for the aforementioned non-structural elements. This method is universally applicable and allows for the presentation of seismic loads that satisfy domestic seismic design standards [20]. Specifically designed for the seismic performance verification of common non-structural elements, this testing method systematically organizes and presents vibration table equipment procedures and methods by referencing design standards such as ASCE 7-16 [21] and testing specifications such as FEMA 461 [22] and IEEE Std 344 [23]. The studies introduced below have evaluated the seismic performance of non-structural elements belonging to the same category as the target structure in this research. These evaluations were conducted through shaking table testing and analytical verification procedures following ICC-ES AC 156. This body of studies served as foundational reference material for conducting the present study.

Zhou (2019) conducted shaking table testing to provide an effective method for realistically evaluating the seismic performance of non-structural components (e.g., ceilings, pipe supports and hangers, and glass screen walls). During shaking table testing, non-structural components are sensitive to the floor motion of the supporting frame to which they are attached. To address the need for accurate measurement methods in response to this sensitivity, a control method was proposed. Following this approach, the dynamic coupling between the supporting frame and the shaking table was overcome. The method allowed for accurate reproduction of response spectra in multiple directions, verifying the seismic performance [24].

Nam (2012) conducted seismic performance evaluations of water-based fire protection systems (general piping, seismic-resistant piping, pumps, etc.) through shaking table tests. To investigate the dynamic response characteristics of the target non-structural components, El-Centro earthquake waves at 50%, 70%, 100%, and 120% amplitude levels were applied. The test results verified, through shaking table testing, that seismic-resistant facilities exhibit better seismic performance compared to conventional facilities, as evidenced by larger displacement and acceleration responses in the conventional facilities. Additionally, by examining the acceleration response spectra at different amplitude levels, it was revealed that even in small-scale earthquakes, destruction of conventional facilities could occur [25].

Maekawa (2010) highlighted that the thin side walls of reservoirs are not very rigid, and seismic-induced vibration modes (slushing and bulging) may influence the distribution of liquid pressure and seismic loads. Therefore, in the seismic design of reservoirs, it is important to theoretically and experimentally consider such elastic deformations. To investigate this, shaking tests were conducted on a scaled-down tank model partially filled with water, to examine the dynamic fluid pressure behavior and seismic safety. The results emphasized that the shape and size of the dynamic fluid pressure distribution vary between positive and negative pressures, and they are influenced by the magnitude of input acceleration [26].

Silvestri (2022) conducted a seismic performance evaluation on a full-scale flat-bottomed steel silo filled with soft wheat, characterized by an aspect ratio of 0.9, using shaking table tests. Various sensors, including accelerometers and pressure cells, were employed to monitor static and dynamic responses to measure seismic force-induced response signals. The test results indicated that the fundamental frequency slightly decreased with increasing acceleration, while it slightly increased with compaction of the granular material. Moreover, for close-to-resonance input, the dynamic amplification (in terms of peak values of accelerations) increased along the height of the silo wall up to values of around 1.4 at the top surface of the solid content. This research argued that the dynamic overpressures appeared to increase with depth—in contrast to the expectations of EN1998-4 [27]—and were proportional to the acceleration [28].

## 3. On-Site Investigation and Specimen Design
### 3.1. Selection of the Target Structure

In this study, a hydrogen storage vessel—characterized by a vertical configuration and composed of metal material among various types of hydrogen storage vessels—was chosen as the target structure. Furthermore, it was restricted to a structure where steel columns were installed on the supporting concrete of a reinforced concrete slab. Field investigations were conducted on actually installed hydrogen storage facilities. However, due to the hydrogen explosion incident that occurred in Gangneung, Republic of Korea, obtaining cooperation from relevant companies and organizations proved to be challenging, limiting our ability to conduct a comprehensive on-site investigation. Additionally, when documenting the results of the on-site investigation, the specific locations of the hydrogen storage vessels could not be disclosed for security reasons. Therefore, the section of this paper detailing the on-site investigation results is presented using regional names only. For these reasons, the study was confined to three specific regions where on-site investigations were feasible: Gangneung, Samcheok, and Ulsan. Following the on-site investigation overview provided in Figure 2, on-site investigations were conducted for each respective structure [29]. During the on-site investigations, we focused on examining the concrete strength, installation configuration, and anchor specifications at the base of the hydrogen storage vessels. The strength of the concrete was measured using Schmidt hammer testing and subsequently applied in the design of experimental specimens and numerical analyses.

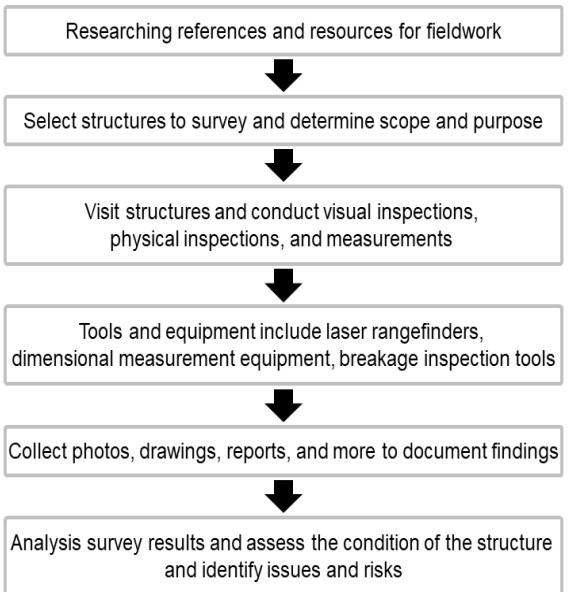

**Figure 2.** Outline of the on-site investigation.

### *3.2. Results of the On-Site Investigation*

This field investigation served as a preliminary investigation to gather essential data for future shaking table tests on vertical hydrogen storage vessels. The primary objective was to investigate the installation conditions, as well as the configuration and dimensions of the foundations of vertical hydrogen storage vessels currently deployed in real-world settings. To achieve this goal, a series of activities were carried out, including visits to representative regions in Korea where vertical hydrogen storage vessels are installed, along with the collection of information through on-site assessments and the examination of relevant documents. Figure 3 compares the support structure types and anchor specifications of the target structures installed in each region. In the case of the target structure installed in Gangneung (as shown in Figure 3a), the column shape for support was constructed with H-beams (300 mm × 300 mm × 10 mm × 15 mm) with a height of 980 mm, and the storage vessel was supported by three of these columns. The height from the top of the storage vessel to the foundation concrete was measured to be 8500 mm, and the size of the supporting concrete was measured as $750 \times 650 \times 440$ mm$^3$. The anchors connecting the structure to the support were D30 pre-installed set anchors, with a total of six anchors, two per support point. In the case of the target structure installed in Samcheok (as shown in Figure 3b), the column shape for support was constructed with L-angle steel (75 mm × 75 mm × 9 mm) and had a height of 965 mm, with a total of four columns supporting the storage vessel. The height from the top of the storage vessel to the concrete was measured as 4000 mm, and the supporting concrete had a panel-shaped single piece with dimensions of $1650 \times 1650 \times 50$ mm$^3$. The anchors connecting the structure to the support were D20 pre-installed set anchors, with a total of four anchors, one per support point. In the case of the target structure installed in Ulsan (as shown in Figure 3c), the column shape for support was constructed with square-shaped steel (220 mm × 220 mm × 10 mm) with a height of 800 mm, and the storage vessel was supported by three of these columns. The height from the top of the storage vessel to the foundation concrete was measured as 8000 mm, and the size of the supporting concrete was $470 \times 440 \times 240$ mm$^3$. The anchors connecting the structure to the support were D30 pre-installed set anchors, with a total of 12 anchors, four per support point. Table 1 compares the measured values of the foundation concrete strength of the target structures installed in each region, as obtained using Schmidt hammer testing [30].

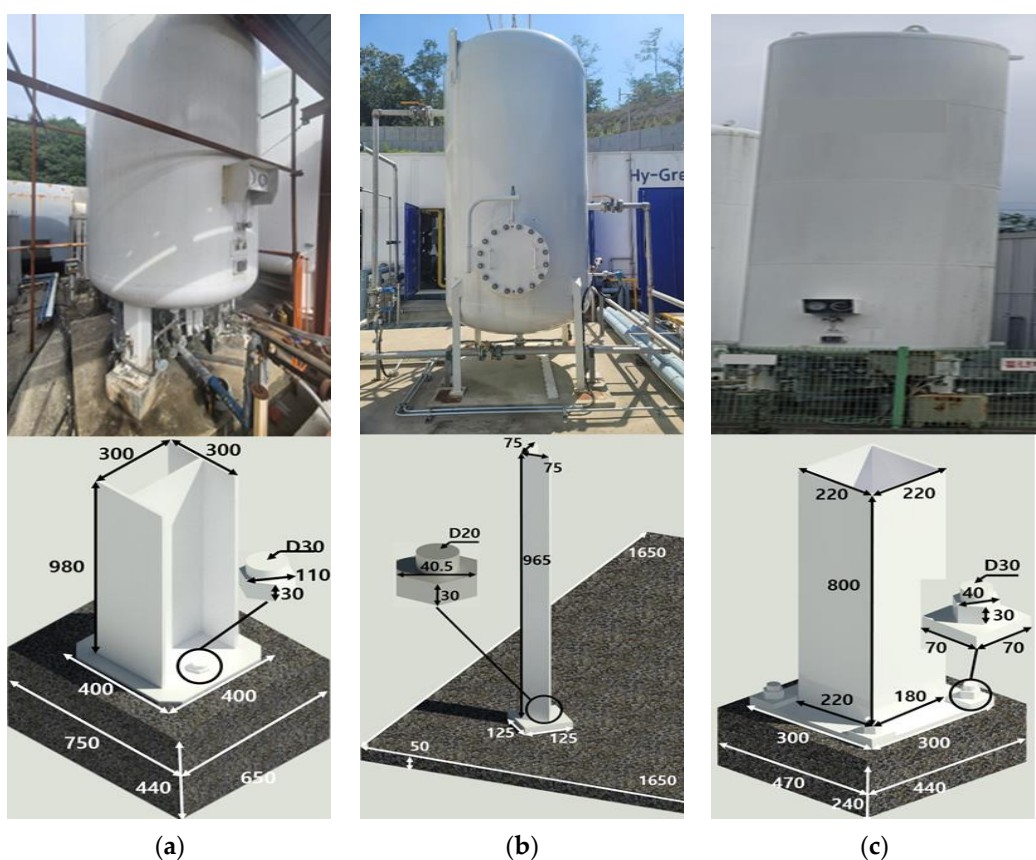

**Figure 3.** The configuration and support system of the target structures installed in each region (unit: mm): (**a**) Gangneung, (**b**) Samcheok, and (**c**) Ulsan.

**Table 1.** Review of the non-destructive strength of the substructure concrete in each region.

| Case No. | Field Investigation Value for Concrete Compressive Strength (MPa) | | | | | |
| | Gangneung | | Samcheok | | Ulsan | |
| | Rebound Hardness | Correction Value | Rebound Hardness | Correction Value | Rebound Hardness | Correction Value |
|---|---|---|---|---|---|---|
| 1 | 40 | 38 | 35 | 35 | 37 | 36 |
| 2 | 36 | 36 | 36 | 36 | 40 | 38 |
| 3 | 42 | 40 | 36 | 36 | 40 | 38 |
| 4 | 39 | 38 | 35 | 35 | 36 | 36 |
| 5 | 38 | 37 | 26 | 28 | 32 | 33 |
| 6 | 44 | 41 | 23 | 26 | 40 | 38 |
| 7 | 41 | 39 | 24 | 27 | 41 | 39 |
| 8 | 42 | 40 | 28 | 30 | 40 | 38 |
| 9 | 40 | 38 | 29 | 31 | 39 | 38 |
| 10 | 41 | 39 | 30 | 31 | 37 | 36 |
| 11 | 42 | 40 | 24 | 27 | 40 | 38 |
| 12 | 40 | 38 | 27 | 29 | 40 | 38 |
| 13 | 43 | 41 | 29 | 31 | N/A | N/A |
| 14 | 43 | 41 | 25 | 28 | 38 | 37 |
| 15 | 39 | 38 | 24 | 27 | 39 | 38 |
| 16 | 39 | 38 | 28 | 30 | 38 | 37 |
| 17 | 41 | 39 | 26 | 28 | 39 | 38 |
| 18 | 43 | 41 | 28 | 30 | 37 | 36 |
| 19 | 39 | 38 | 29 | 31 | 37 | 36 |
| 20 | 45 | 42 | 26 | 28 | 38 | 37 |
| AVG. | N/A | 39 | N/A | 30 | N/A | 37 |
| SD | N/A | 1.5 | N/A | 2.9 | N/A | 1.4 |
| CV | N/A | 3.8 | N/A | 9.6 | N/A | 3.8 |

The Schmidt hammer test is one of the prominent non-destructive testing methods used to estimate the compressive strength of concrete, by measuring its surface hardness. This test relies on experimental observations indicating a specific correlation between the rebound value (R) obtained by impacting the hardened concrete surface with a Schmidt hammer and the compressive strength of the concrete. When striking the hardened surface of concrete with a Schmidt hammer, the resulting rebound value represents the internal rebound energy, and the compressive strength of the concrete is estimated based on the magnitude of this rebound value. For the selected measurement locations determined by visual inspection results, surface smoothing operations were performed. The impact points were standardized at 3 cm intervals, forming a grid of 20 points where lines were drawn at intervals of five rows and four columns. The impact tests were conducted with an N-type Schmidt hammer in a vertically downward direction ($\alpha = -90$ degree), and adjustments for the slope angle were considered when estimating the strength. According to the RILEM Recommendations [31], for the determination of the rebound hardness (R), values outside the range of each rebound hardness measurement were excluded. The remaining values were then used to calculate the arithmetic mean rebound hardness value (R). If four or more rebound values were excluded, the rebound hardness values (R) for that measurement point were discarded. A measured rebound value showing significant variability is generally indicative of non-uniformity in the quality of the concrete. To correct for the impact direction, measurements were conducted in the horizontal and vertical direction to ensure the most stable readings of the rebound hardness. Meanwhile, in Korea, a proposed formula for estimating the compressive strength ($F_c$) of concrete from the corrected rebound hardness ($R_0$) is widely utilized, referencing Equation (1) proposed by the Architectural Institute of Japan [32]:

$$F_c = (7.3R_0 + 100) \times 0.098 (\text{Mpa}) \tag{1}$$

The compiled data for each region are presented in Table 1. The measurement results indicated that the average strength of the concrete in Samcheok was 19–23% lower than in other surveyed regions.

### 3.3. Design of the Test Specimen

The primary objective of this paper is to assess the seismic performance of vertical hydrogen storage vessels during an earthquake, using shaking table testing. To achieve this goal, a hydrogen storage vessel design, as depicted in Figure 4 and Table 2, was developed based on site survey data and seismic design standards. The configuration of the selected structure, including the hydrogen storage vessel and supporting columns, was determined by identifying the facility in Samcheok, where the L-shaped steel structure supporting the hydrogen storage vessel is considered to have the weakest rigidity among the three surveyed locations. This choice was made via a risk assessment. The quantity and specifications of the anchors were determined in accordance with the KGS FU671 standards [33], which outlines the Technology and Inspection Standards for Hydrogen Fuel Facilities. The standards specify M20 anchors for equipment with a weight of 1 ton, requiring four units and a thread length of 250 mm when M20 anchors are utilized. This information, as specified in the standards, was also consistent with the on-site survey, leading to our decision to adopt these parameters. The strength of the foundation and protrusion concrete was determined to be the lowest measured value of 30 Mpa, through on-site surveys. Specifications for the foundation and protruding concrete were determined based on anchor embedment length and on-site survey data. Consideration was given to conditions that would allow for installation during future shaking table testing.

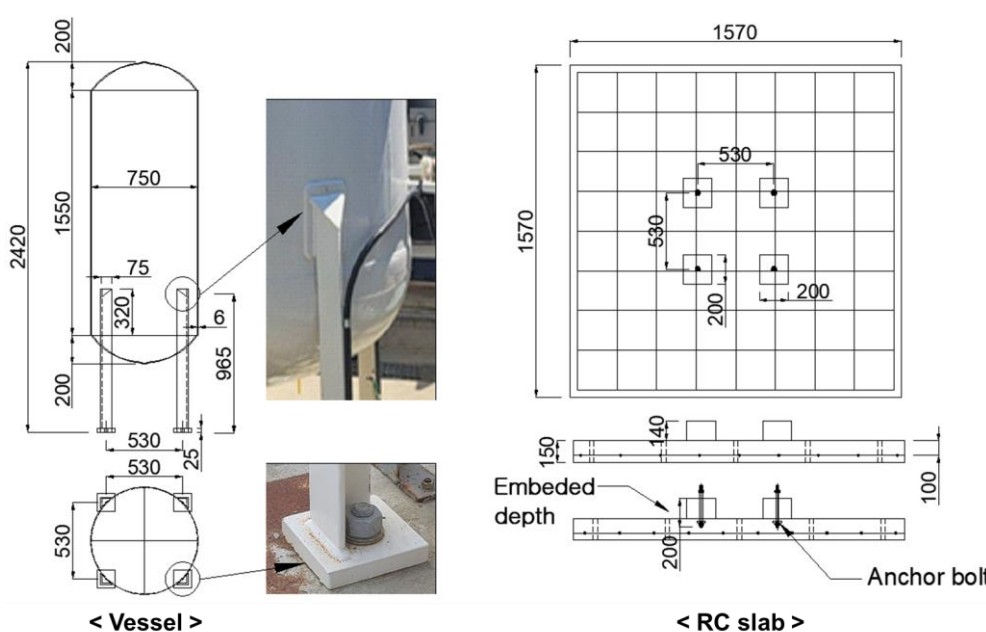

**Figure 4.** Summary diagram of specimen design for shaking table testing (unit: mm).

**Table 2.** Specifications of the test specimen.

| Type | Weight (kg) | Size (mm) | | | | Description |
|------|------|------|------|------|------|------|
| | | L | W | H | t | |
| Vessel | 1046 | 530 | 530 | 2420 | 6 | SS400 L-shape angle (75 × 75 × 9.5) |
| RC slab | 908 | 1570 | 1570 | 290 | 150 | $f_{ck}$: 30 Mpa Protrusion concrete (200 × 200 × 140) |
| Anchor | - | 250 | - | - | - | ASTM A36 [34], M20 Embedment depth: 200 |

## 4. Numerical Verification for Design

### 4.1. 3D Analysis Modeling

In this chapter, the aim was to validate the reliability of test results by numerically analyzing the seismic performance of the hydrogen storage vessel through the construction of a 3D dynamic analysis model. To achieve this, the commercial structural analysis software ABAQUS was used. In the finite element analysis (FEA) modeling, time history analysis—one of the prominent numerical analysis methods used to understand the dynamic performance of structures—was employed to examine the seismic behavior. Typically, dynamic behavior is more accurate than static behavior, and nonlinear behavior is more accurate than linear behavior for structures. Time history analysis is generally the most accurate method for understanding the nonlinear dynamic behavior of structures and is commonly used for evaluating the seismic performance of target structures. Furthermore, it can provide more objective and accurate results since it can capture dynamic behavior characteristics such as relative displacements, anchor load transfers, and acceleration responses, which cannot be determined through static analysis or response spectrum analysis [35]. The configuration of the Finite Element (FE) modeling, as shown in Figure 5, consisted of the hydrogen storage vessel, support columns, and foundation concrete, using three degrees of freedom at eight nodes and minimal integration points with 8-Node Solid elements (C3D8R). Additionally, pre-installed set anchors, used to secure the structure to the concrete, were also implemented using C3D8R elements. The FE model consisted of a total of 295,643 nodes and 222,556 elements. The mesh size for the connection points between the main body and support columns—where stress concentration was anticipated—was set to 10 mm × 10 mm. For anchors and the lower plates of support columns, a mesh size

of 5 mm × 5 mm was utilized. Mesh configurations for areas other than vulnerable points were appropriately designed to facilitate convergence time during analysis.

To represent the areas of the target structure used in the shaking table test that were connected by welding in the analysis, the welding effect was implemented by combining the Tie function and Coupling function of the ABAQUS platform. This involved constraining degrees of freedom according to the coupling direction, based on translational and rotational degrees of freedom. To model the contact surfaces between two elements with different material properties that can induce sliding effects considering friction, the Interaction feature, which allows for interaction within the analysis program, was employed [36].

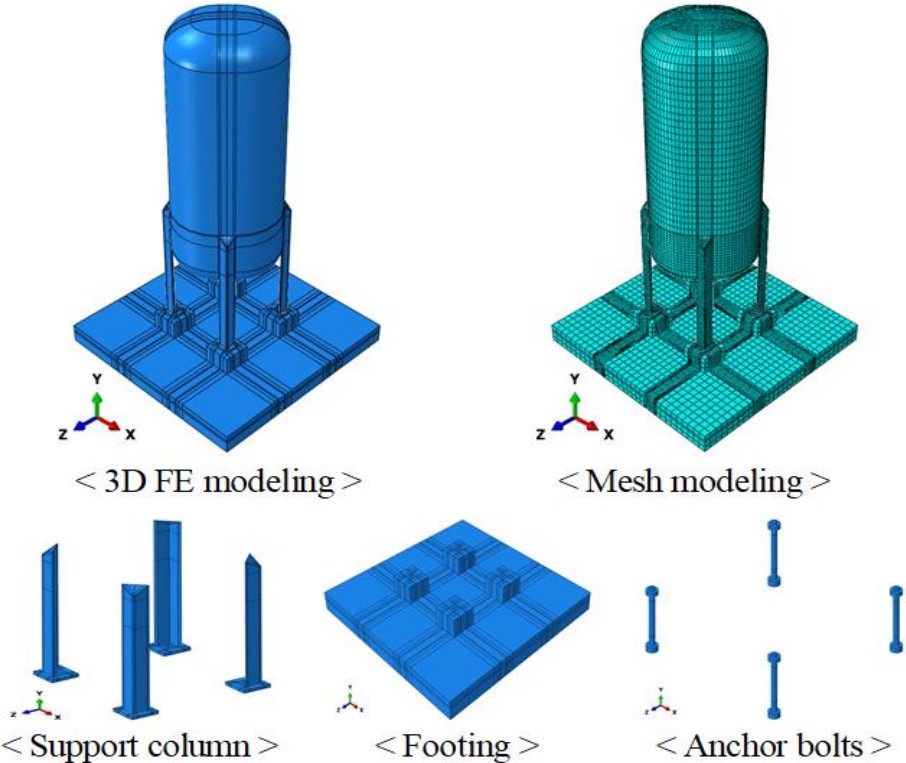

**Figure 5.** Various components of FE modeling.

To model the boundary conditions of the anchor (as shown in Figure 6), the contacting surface between areas where gaps (such as between the anchor and nut) are not anticipated, is simulated using a monolithic (e.g., tied surfaces in ABAQUS). The mechanical interaction between the anchor and steel plate and between the anchor and concrete surface is modeled to prevent penetration using friction formulations in the tangential direction and hard contact in the normal direction. The penalty method was applied, by setting the friction coefficient at 0.1 for steel-to-steel interactions and 0.43 for steel-to-concrete interactions in the tangential direction [37,38]. The hard contact option was employed for normal interaction, allowing separation after contact in the interaction model. The numerical value used as the friction coefficient between the two abovementioned material surfaces was based on previous research, which derived the average from a parametric study investigating the design and behavior of components (such as anchors) used to transmit forces from the upper steel structure to the supporting concrete foundation [39].

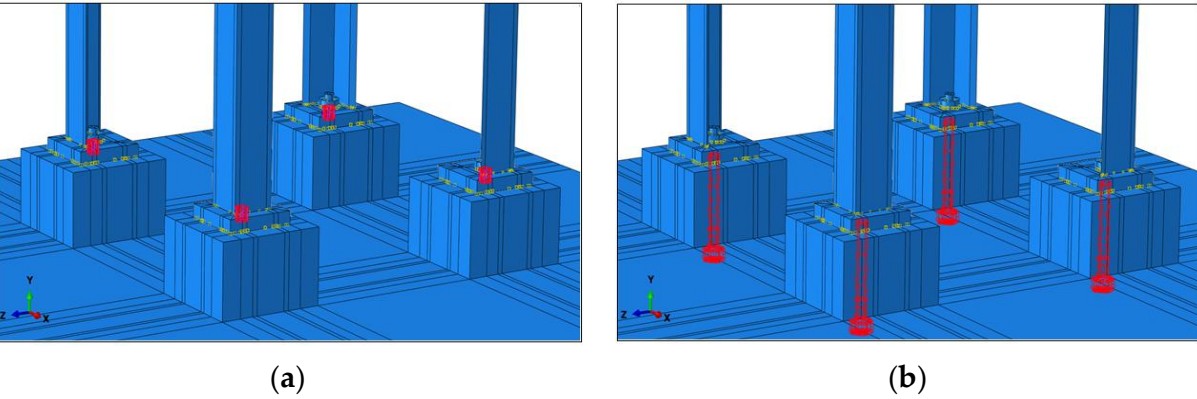

|(**a**)|(**b**)|

**Figure 6.** Friction coefficient of the contact for each material: (**a**) steel-to-steel and (**b**) steel-to-concrete.

### 4.2. Material Modeling

Concrete is a composite material with unique properties, exhibiting anisotropic and nonlinear behavior. Therefore, material models that account for these characteristics should be applied. To depict the concrete damage behavior in numerical analysis, a model incorporating both compressive failure and tensile cracking (as shown in Figure 7) was employed. For this purpose, the Concrete Damaged Plasticity model provided in ABAQUS 6.20 (2020) was applied, following the approach of previous research related to concrete material modeling [40]. The concrete material properties were incorporated based on the Simplified Damage Plasticity Model (SDPM) to capture compression behavior [41]. Additionally, a linear model that can capture the tensile behavior of the anchor was considered. SDPM defines the compression behavior based on an elastic modulus of 24,830 MPa and a Poisson's ratio of 0.2, with a reference compressive strength of 30.0 Mpa for concrete specimens. Furthermore, the tensile strength was set to 15% of the compressive strength, and to enhance the convergence of the analysis results, a 1-linear strain corresponding to 1% of the yield strength was considered for tensile cracking deformation [42]. The mechanical properties of the M20 anchor were determined by the bolt grade, following the F8T standard [43]. The yield strength and tensile strength were 640 Mpa and 800 Mpa, respectively. The elastic modulus was 200,000 Mpa, and the elongation was 12%. Furthermore, the model was constructed based on a 2-linear model allowing for the yielding deformation of the anchor during load transfer [44].

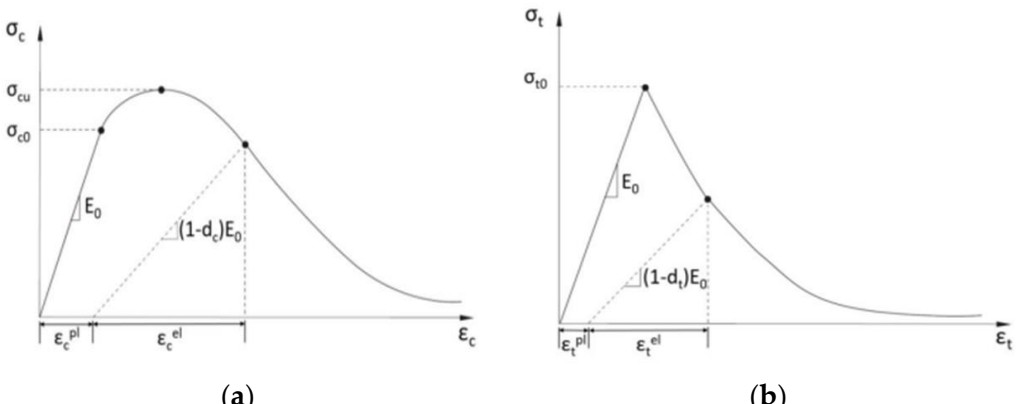

|(**a**)|(**b**)|

**Figure 7.** Response of concrete to a uniaxial loading condition: (**a**) compression behavior and (**b**) tension behavior.

### 4.3. Seismic Input and Boundary Conditions

The seismic input applied in the numerical analysis corresponds to an artificial seismic wave based on the KDS criteria, specifically the AC 156 Amp. 100%. The seismic excitation was applied unidirectionally along the axis with an amplitude, as represented in Figure 8. The seismic input direction for structural analysis was specified along the X-axis, which represents the dominant response direction for the target structure. This determination was made based on prior studies and literature. Before performing time history analysis, eigenvalue analysis must be performed to predict the unique vibration of the target structure. Through this analysis, it is possible to determine the participation factor of the mass in the model (with a requirement of 90% or more participation for satisfaction). This ensures confidence in the reliability of the results obtained from the subsequent time history analysis. For the eigenvalue analysis, all directions of the foundation concrete were constrained (All-fixed), and for the time-history analysis, depending on the results of the eigenvalue analysis, acceleration was applied and constraints were released in the relatively vulnerable uniaxial direction (Axis-X). The reason for employing this analysis approach is that the program used for the analysis imposes constraints that prevent sharing the same node for velocity, displacement, and acceleration simultaneously in the analysis step.

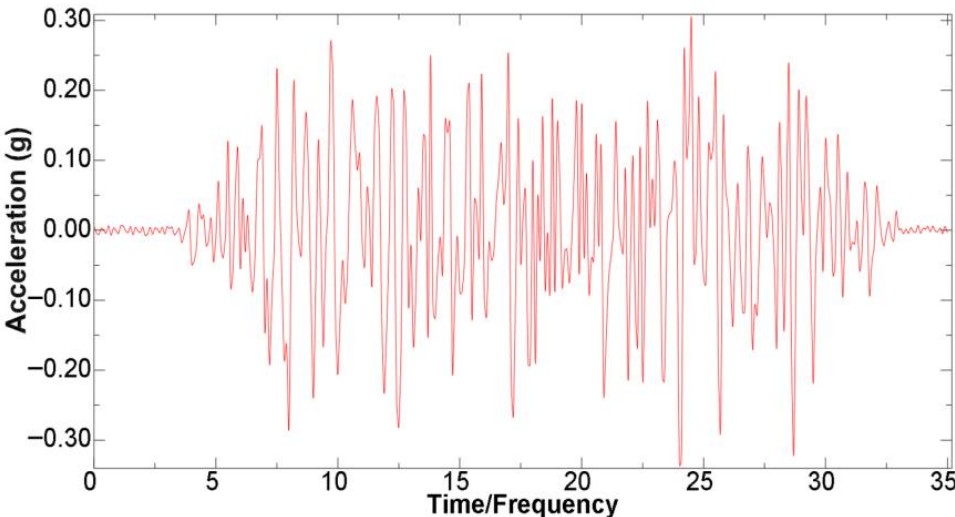

**Figure 8.** Seismic input applied in the numerical analysis (AC 156 Amp. 100%).

### 4.4. Results Analysis

#### 4.4.1. Natural Frequency

For the natural frequencies extracted from the analysis, the Lanczos Method was employed for modal extraction, and natural frequencies for a total of 10 modes were measured. As outlined in Table 3, the value of 14.299 Hz was determined for the first mode, where more than 90% of the mass participated. This value was chosen as the representative natural frequency for the primary mode in the numerical analysis. The reason for the identical natural frequencies in the first and second modes can be deemed reasonable, as the target structure is symmetrically configured. The abovementioned Lanczos Method is an iterative power method devised by Cornelius Lanczos for finding eigenvalues and eigenvectors of a Hermitian matrix with respect to frequency/time. This method is applied to matrices represented by n, m, and it is specifically employed within the eigenvalue analysis feature in ABAQUS. The relevant equations and procedures for its application can be found in the ABAQUS Theory Manual, specifically in the "2.5.1 Eigenvalue Extraction" section. The Lanczos Method is efficient for extracting eigenvalues by utilizing the characteristics of the matrix, particularly in the context of symmetric matrices. Figure 9 illustrates the mode shape for the abovementioned first mode.

**Table 3.** Eigenvalue analysis results.

| Mode No. | Eigenvalue | Natural Frequency (cycle/sec) | Effective Mass * (X-Component) |
|---|---|---|---|
| 1 | 8071.7 | 14.299 | 1.05935 |
| 2 | 8072.1 | 14.299 | 0.000339 |
| 3 | 34,610 | 29.609 | $1.24 \times 10^{-9}$ |
| 4 | 57,194 | 38.062 | $6.28 \times 10^{-8}$ |
| 5 | 57,194 | 38.062 | $6.01 \times 10^{-8}$ |
| 6 | 57,194 | 38.062 | $6.42 \times 10^{-8}$ |
| 7 | 57,195 | 38.063 | $6.12 \times 10^{-8}$ |
| 8 | 59,922 | 38.959 | $1.88 \times 10^{-6}$ |
| 9 | 59,923 | 38.960 | $1.66 \times 10^{-6}$ |
| 10 | 59,923 | 38.960 | $2.09 \times 10^{-6}$ |

* The total effective mass at axis-X is 1.05955.

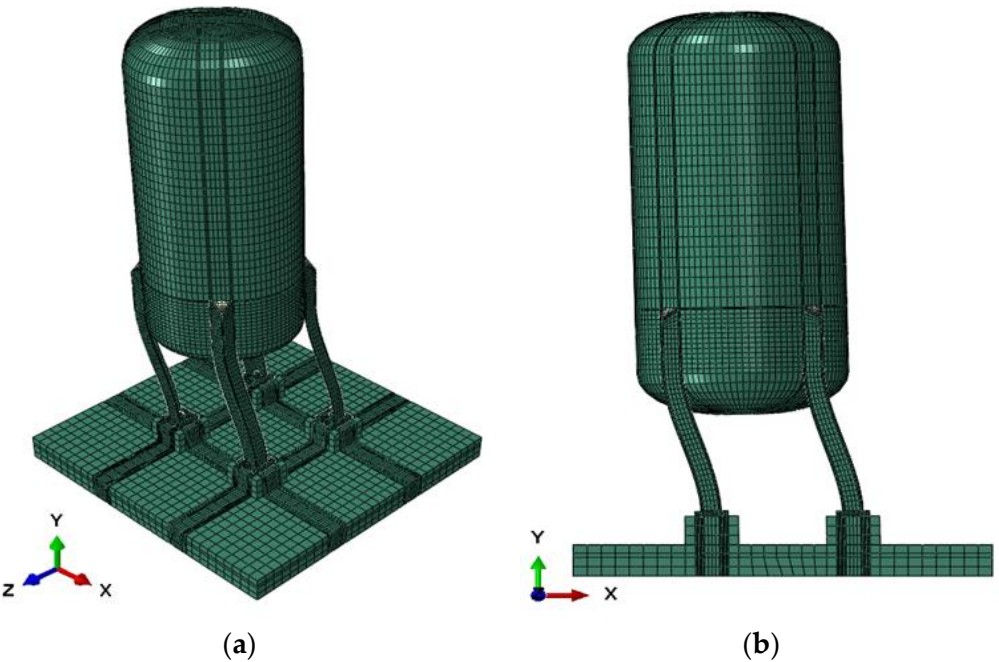

| (a) | (b) |

**Figure 9.** View of the first mode shape (scale: 1:200): (**a**) isometric and (**b**) axis X-Y: side.

### 4.4.2. Acceleration Response

The acceleration response and peak values obtained through time history analysis (based on the input motion for each location under seismic loading) are presented in Figure 10. The maximum acceleration response occurred at the location corresponding to the footing concrete (A1) of the target structure. It can be observed that the acceleration response decreases as it moves towards the upper part of the hydrogen storage vessel (A4). Accordingly, the amplification generated at the lower part by seismic motion is predicted to decrease as it is transferred towards the point where the center of gravity of the target structure is located, resulting in a relatively smaller impact on acceleration. However, in the case of support columns composed of flexible materials such as steel, acceleration amplification may be induced. This is anticipated to have a significant impact on the anchor system connecting the hydrogen storage vessel and concrete.

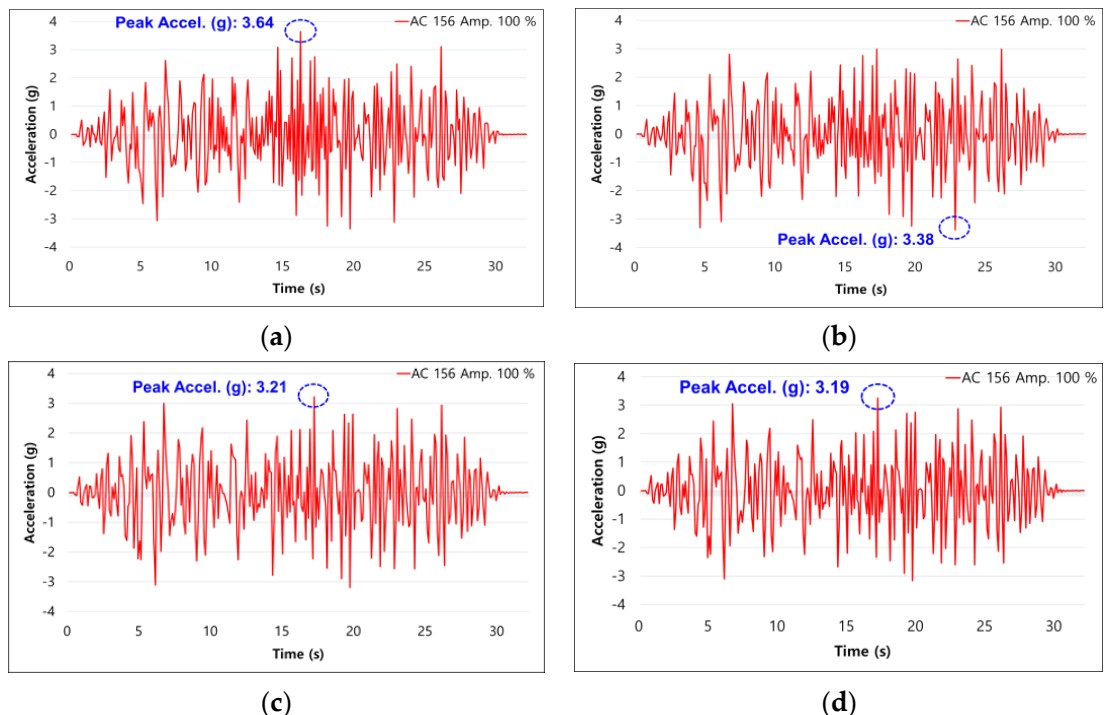

**Figure 10.** Acceleration response by location of target structure according to seismic load: (**a**) footing concrete (A1), (**b**) support column (A2), (**c**) middle of the hydrogen storage vessel (A3), and (**d**) top of the hydrogen storage vessel (A4).

4.4.3. Stress Response

It was observed that the amplification of acceleration is intensified depending on the form and material of the support system of the target structure. Accordingly, the location most affected by this was selected, and the stress generated in the structure was reviewed, as shown in Figure 11. The review locations were determined to be three areas where stress concentration occurs: the connection point between the vessel body and support column (S1), the bottom plate of the support column (S2), and the anchor (S3). When examining the stress in isotropic materials such as steel, Von Mises stress was investigated. At location S1, which is the welded connection between the vessel body and the support column, the maximum stress was observed to be 64.30 MPa, the highest among the three measured locations. The maximum stress for the bottom plate of the support column and the anchor were relatively similar, measuring 41.33 MPa and 42.09 MPa, respectively. As a result, the maximum stresses observed at all measured locations are below the yield strength of the common structural steel SS 400, which is 225 MPa. Therefore, it is concluded that the structural integrity of the target structure is ensured.

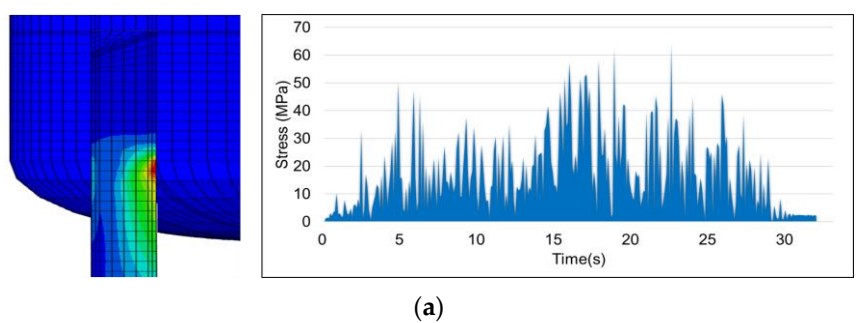

(**a**)

**Figure 11.** *Cont*.

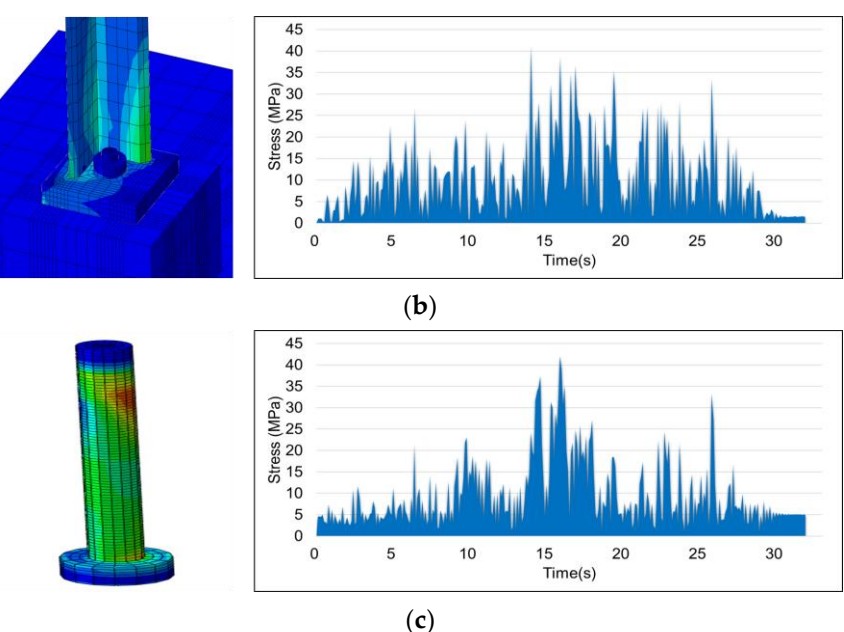

(**b**)

(**c**)

**Figure 11.** Results of comparison of areas where stress is concentrated: (**a**) connecting point (S1), (**b**) support column (S2), and (**c**) anchor bolts (S3).

### 4.4.4. Review of the Safety of Protrusion Concrete

In the case of the vertical hydrogen storage vessel reviewed in this paper, the unique characteristics of the structure, supported by a high center of gravity and anchors, may result in concentrated complex loads such as moment loads on the protrusion concrete during seismic loading. Accordingly, in Figure 12, where the anchor stress is most significant in the protrusion concrete due to the seismic loading, the splitting tensile strength is assessed at the anchor insertion area (C1), and the bending tensile stress is evaluated at the joint area connected to the slab (C2) to confirm structural integrity.

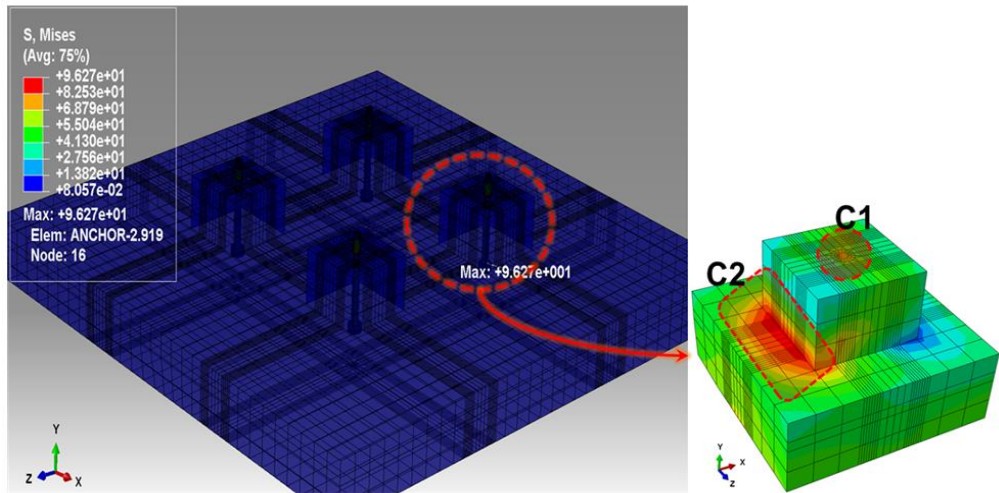

**Figure 12.** Review area of protrusion concrete according to seismic loading.

The permissible splitting tensile strength and bending tensile stress for the reviewed protrusion concrete area were calculated based on Equations (2) and (3) [45,46]. These values were used as references during the review process to assess the structural integrity:

$$f_{sp} = 0.57\lambda\sqrt{f_{ck}} \tag{2}$$

$$f_r = 0.63\lambda\sqrt{f_{ck}} \qquad (3)$$

Here, the coefficient "λ" is used when lightweight concrete is cast, while for the case where regular ordinary concrete is used, the concrete coefficient was set to 1, considering the actual field installation conditions with typical aggregates. The maximum splitting tensile strength occurring in the upper part (C1) of the protrusion concrete, where direct contact with the anchor for the structural support is established, was measured as 2.87 MPa, as shown in Figure 13a. This indicates a safety margin of approximately 8%, considering the allowable splitting tensile strength of 3.12 MPa. The maximum bending tensile stress occurring in the joint area (C2) between the protrusion concrete and the foundation slab concrete was measured as 3.17 MPa, as shown in Figure 13b. This indicates a safety margin of approximately 8%, considering the allowable bending tensile stress of 3.45 MPa.

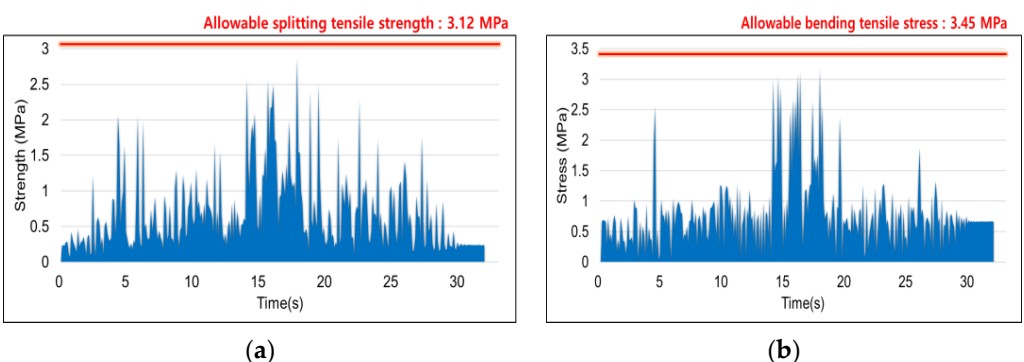

(**a**) (**b**)

**Figure 13.** Safety review of the protrusion concrete: (**a**) upper part (C1) and (**b**) connecting part (C2).

Accordingly, it is determined that the structural integrity for the AC 156 Amp. 100% artificial earthquake, satisfying the seismic design criteria for buildings, has been secured. However, it is predicted that issues may arise when applying more stringent seismic design criteria, as the safety margin with respect to the structure's own safety factor is relatively limited.

## 5. Shaking Table Test

### 5.1. Specimen Design and Fabrication

The specimens for the shaking table testing of the vertical hydrogen storage vessel were designed and fabricated based on on-site surveys and numerical studies. The experimental conditions for the shaking table testing, as outlined in Section 3.3 of this paper, were taken into account during the design and fabrication process. To configure the support structure with four anchors, it was decided to exceed 1 ton, according to the KGS FU671 standards. The shape of the specimen was determined by assuming the weight under conditions in which it was filled with water, considering the actual size, experimental conditions, and stability during the test. The thickness of the body was selected based on analytical verification results. The shape and thickness of the body and supporting columns were selected based on analytical verification results. For producing the footing system, anchor bolts were considered with M20 specifications made of ASTM A36 material, incorporating studs and utilizing the cast-in-place installation condition. Embedment depth, foundation fixation methods, and other details were determined based on on-site surveys and design standards. The form of the anchor section of the test specimen and the fixed foundation section was chosen to be protrusion concrete without reinforcement, and the specifications of the protrusion concrete were produced considering design criteria and the shear span for concrete.

### 5.2. Input Seismic Motion

The Required Response Spectrum (RRS) for the input seismic motion used in shaking table testing was developed based on the seismic design criteria, which are more stringent than the KDS criteria applied in the analytical study. The RRS was created by referencing AC 156 with an amplification of 150%, following common application criteria for seismic design. The reason for applying more stringent seismic performance criteria is to experimentally validate the aspects predicted through the analytical results using methods such as shaking table testing, to ensure practical verification. The acceleration ($S_{DS}$) for the development of the RRS was calculated according to Equations (4) and (5):

$$S = Z \times I \tag{4}$$

$$S_{DS} = S \times 2.5 \times F_a \tag{5}$$

The seismic zone factor ($Z$) for the determination of effective horizontal ground acceleration ($S$) was assumed to be 0.11 g, encompassing the majority of domestic regions. The hydrogen storage vessel, the subject of the test, was classified as earthquake-resistant special grade due to its potential significant impact on society in the event of a seismic disaster and substantial functional impairment. Therefore, with a return period of 2400 years, the hazard coefficient ($I$) was assumed to be 2.0, resulting in $S = 0.22$ g. The short-period ground amplification factor ($F_a$) was assumed to be 1.5 according to KDS, considering the assumption of soft soil and effective horizontal ground acceleration. Therefore, $S_{DS} = 0.825$ g. The calculated spectral displacement ($S_{DS}$) was used to present the RRS and seismic parameters according to AC 156 in Figure 14 and Table 4. The Test Response Spectrum (TRS) was required to adhere to the Required Response Spectrum (RRS) within the allowable range of $-10\%$ to $+50\%$ of the RRS. If the TRS exceeded $+50\%$ of the RRS, approval from the verification authority was required.

The acceleration time histories for the seismic simulation test were generated based on the developed RRS, and the damping ratio was set to 5%. Referring to IEEE 344, the correlation function values were adjusted to be below 0.3, and the vibration duration and earthquake duration were set to 30 s and 20 s, respectively, within the frequency range of 0.5 Hz to 50 Hz.

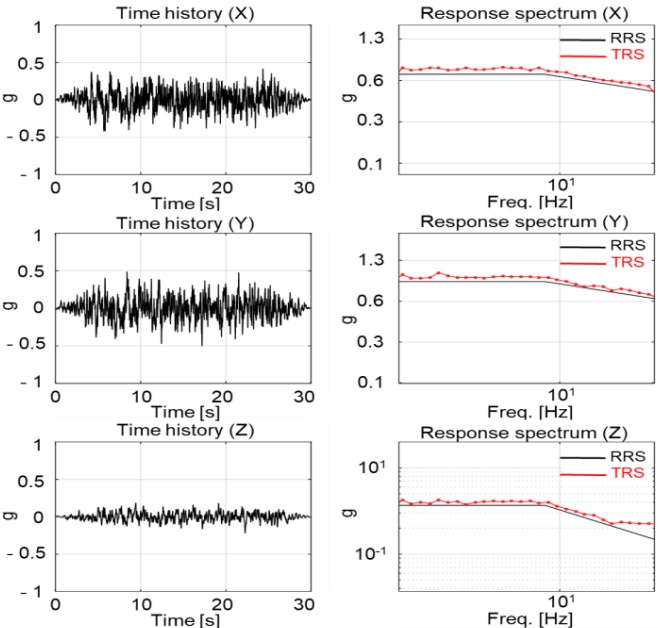

**Figure 14.** Required Response Spectrum (RRS) and Test Response Spectrum (TRS) based on ICC-ES AC 156 with Amp. 150%.

**Table 4.** Seismic parameters of AC 156 based on common application of seismic design criteria.

| Code | $S_{DS}$ (g) | z/h * | $A_{FLEX-H}$ (g) | $A_{RIG-H}$ (g) | $A_{FLEX-V}$ (g) | $A_{RIG-V}$ (g) |
|---|---|---|---|---|---|---|
| Common application of seismic design criteria | 0.825 | 1 | 1.32 | 0.99 | 0.55 | 0.22 |

* z/h denotes the ratio of the location of the structure and non-structural elements.

### 5.3. Test Method and Procedure

The test was conducted at the Earthquake and Disaster Prevention Research Center affiliated with Pusan National University, which possesses the largest shaking table in the Republic of Korea. The specifications and standards of the shaking table are presented in Table 5. The experiment utilized a six-degrees-of-freedom shaking table for seismic simulation testing. To ensure the adequacy of the applied seismic input, the reference accelerometers were installed on the shaking table at locations adjacent to the support structure of the tested facility. Additionally, vibration-response-monitoring instruments were used at key points reflecting the resonant frequencies associated with the tested facility's dynamic characteristics. During this process, measurement instruments were typically installed at appropriate locations agreed upon by the manufacturer, end-user, and testing organization, to assess the seismic performance of the tested facility.

The connection between the RC slab—installed with cast-in-place anchor bolts—and the test specimen body was fixed with the same fastening force of 140 N·m, based on the tightening torque criteria of the anchor bolts. To measure the deformation and stress of the test specimen anchor section in response to the input seismic motion, three-axis strain gauges were attached at the bottom of the supporting column, as shown in Figure 15. Three-axis acceleration sensors were attached to the top, at the height of the center of gravity of the test specimen and supporting column, to measure the acceleration response. Additionally, three-axis acceleration sensors were attached to the shaking table floor, to calculate the acceleration response transfer function for each position of the input acceleration.

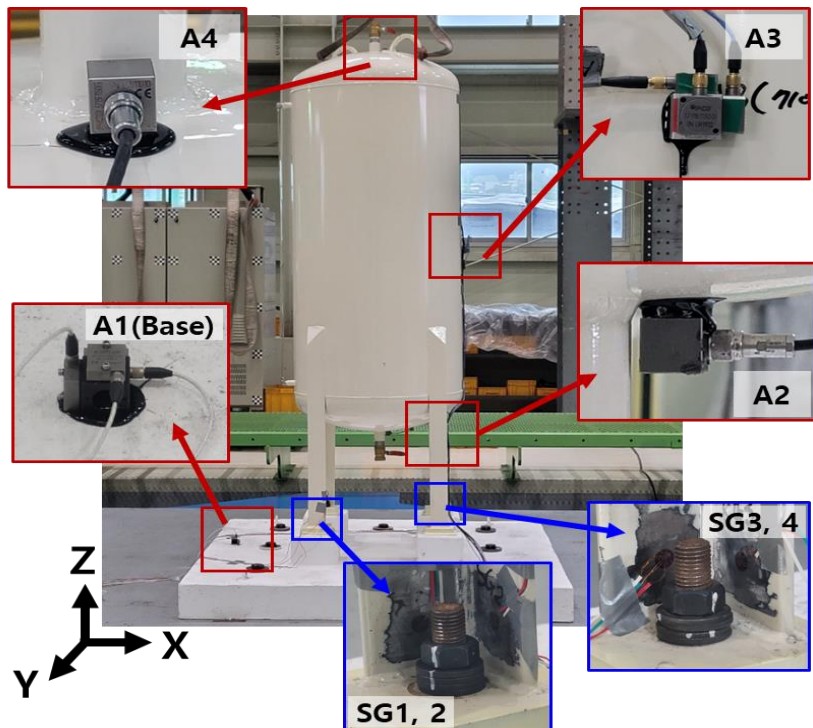

**Figure 15.** Test set-up and sensor location.

**Table 5.** Shaking table specifications.

| Category | Specifications |
|---|---|
| Size | 4 m × 4 m |
| Maximum payload | 30 tons |
| Frequency range | 0.1–60 Hz |
| Maximum stroke | Horizontal: ±300 mm, ±200 mm Vertical: ±150 mm |
| Maximum acceleration | ±3 g (bare table condition) |

The procedure and method of the test are outlined in Table 6. Prior to the seismic simulation test, a visual inspection of the anchor section and a resonance search test were conducted. The resonance search test was conducted using a random wave in the range of 0.5 to 50.0 Hz for each axis direction (X, Y, Z) independently for more than 60 s. Subsequently, the generated acceleration time histories were applied simultaneously in all three axes as input seismic motion for the seismic simulation test. After the seismic simulation test, the test proceeded with the resonance search test and visual inspection, to confirm the changes in dynamic behavior and anchor section damage of the test specimen.

**Table 6.** Test procedure and method.

| Procedure | Method |
|---|---|
| Inspection | Visual inspection |
| Pre-resonance search test | Random wave (amplitude: 0.05 g)<br>Uniaxial motion (X, Y, Z, respectively)<br>Time duration: 60 s<br>Frequency range: 0.5–50.0 Hz |
| Seismic simulation test | Cross-correlation function < 0.3<br>3-axis simultaneous motion<br>Time duration: 30 s<br>Strong motion duration: 20 s<br>Frequency range: 0.5–50.0 Hz |
| Post-resonance search test | Same as pre-resonance search test |
| Inspection | Visual inspection |

*5.4. Test Results*

5.4.1. Resonance Frequencies and Damage Modes

The resonance frequencies were determined by calculating the transfer function ($T_{ab}$) of the response acceleration at each measurement position of the test specimen for the acceleration locations measured on the shaking table. The transfer function is given by Equation (6), where $P_{aa}$ is the power spectral density (PSD) of the input signal, and $P_{ba}$ is the cross-power spectral density (CSD) between the input and output signals:

$$T_{ab}(f) = \frac{P_{ba}(f)}{P_{aa}(f)} \tag{6}$$

The resonance frequencies for the horizontal directions (X, Y) were 14.25 Hz and 13.25 Hz, respectively. Since the test specimen is symmetric in the X and Y directions, the resonance frequencies should ideally be the same as the first and second mode results from the eigenvalue analysis mentioned in the analytical verification. However, due to various factors—such as manufacturing errors in the anchorage part and foundation concrete flatness as well as installation errors caused by bolt fastening—the actual boundary conditions may differ from those applied in the analytical model.

According to the input seismic motion based on the common application criteria of the seismic design standards (as shown in Figure 16), the specimen was damaged during the shaking table test. The failure mode exhibited by the specimen is consistent with the

analysis results, indicating damage due to stress concentration in the settlement area caused by external forces and seismic loads, similar to cases of protrusion concrete damage and stress concentration at the boundary of protrusion concrete observed in the analysis. The concrete along the protrusion concrete area and the boundary with the foundation concrete was damaged, resulting in complete separation of the two structures. Due to this condition, further modal analysis through additional resonance search tests was not feasible.

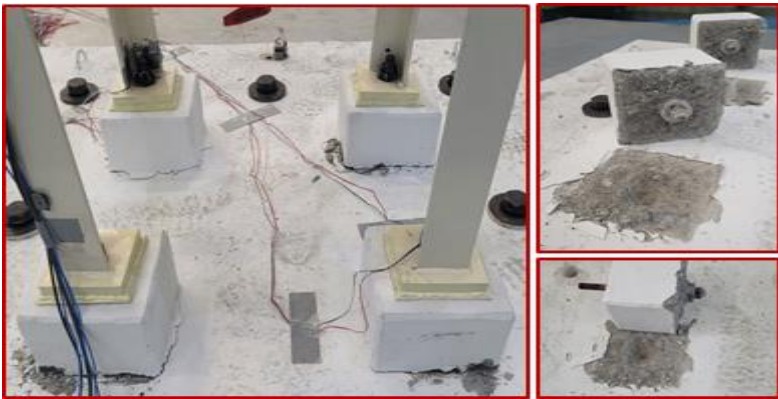

**Figure 16.** Failure mode of the test specimen.

### 5.4.2. Acceleration Response

As shown in Figure 17, sensors were installed at key locations (A1 to A4) to measure the transmitted acceleration and analyze the response behavior of the test specimen to the input seismic load. The measured accelerations were compared with the results of analytical verification to confirm the dynamic behavior pattern. The shaking table test involved conducting seismic simulations in three axes for the test specimen. However, in the analytical verification, only the X-axis was considered. Therefore, the comparison results were limited to the X-axis. The measured acceleration response in the vibration table test showed a pattern where vibration started at the support column and was amplified as it was transmitted to the main body. Then, after approximately 28 s, the protrusion concrete was damaged, leading to the termination of the test. The impact signal due to the damage of the protrusion concrete was measured at the moment when the holding time of the input seismic motion ended, making an accurate comparison challenging. However, the acceleration patterns before 28 s in the shaking table test results and the analytical verification results appear similar, indicating a reasonable outcome. Consequently, if no damage occurs at the support, structural integrity can be assured. However, in the event of damage to the support, the intrinsic impact load of the structure could be added, potentially altering the seismic behavioral characteristics.

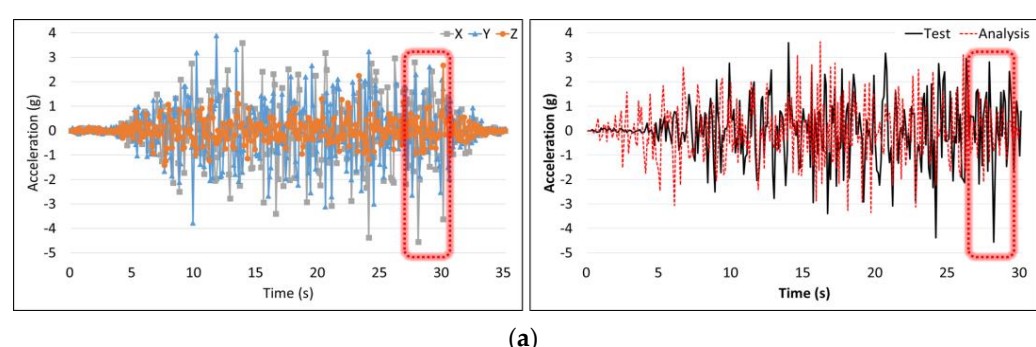

(**a**)

**Figure 17.** *Cont.*

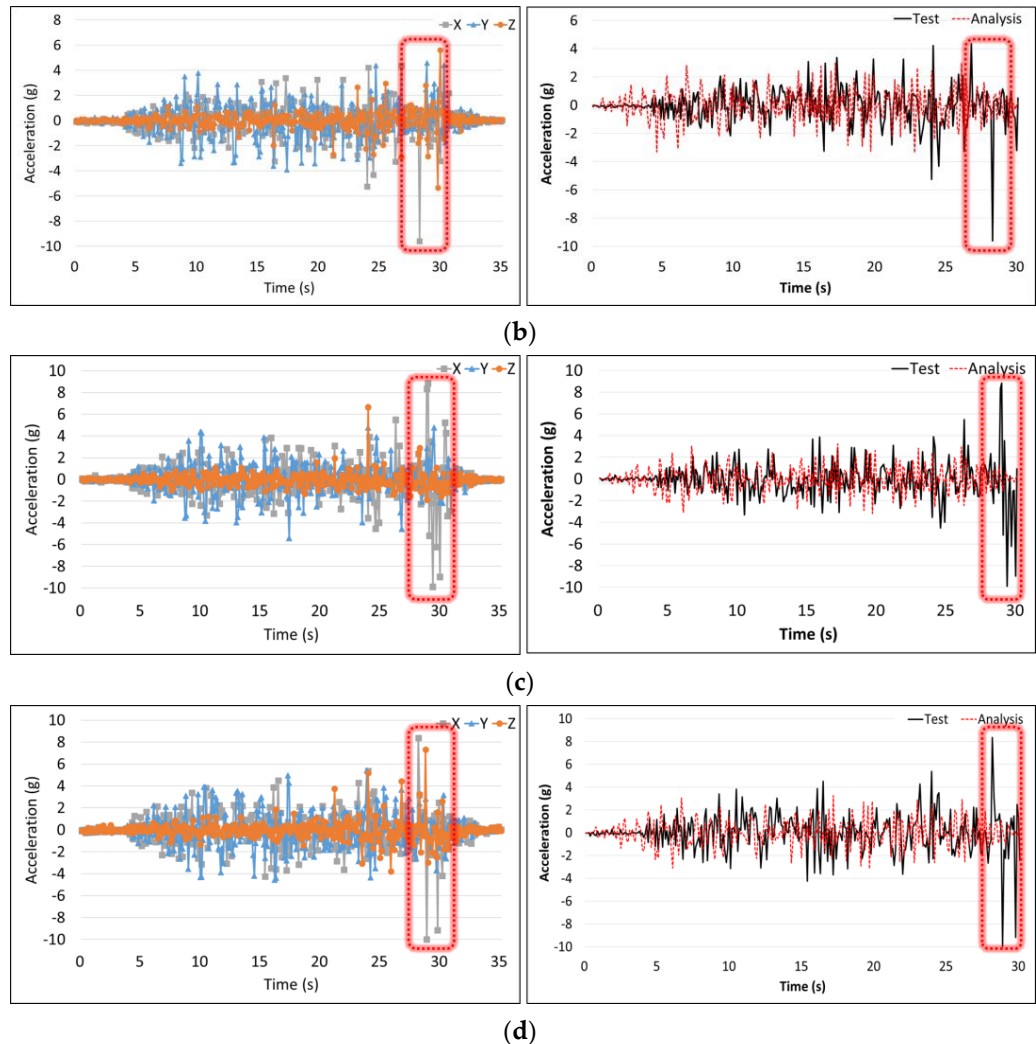

**Figure 17.** Acceleration response history at each location based on shaking table testing and comparison between tests and analytical results: (**a**) footing concrete (A1), (**b**) support column (A2), (**c**) middle of the hydrogen storage vessel (A3), and (**d**) top of the hydrogen storage vessel (A4).

5.4.3. Stress Response

To check the structural integrity of the test specimen, four strain gauges were installed at the anchor area connecting the lower part of the support column and the upper part of the protrusion concrete (where external loads such as earthquakes can be most concentrated), and the stress was measured. During the seismic simulation test, excessive relative displacement occurred between the protrusion concrete and the test specimen, leading to damage in the protrusion concrete. As a result, it separated from the foundation concrete, causing strain gauges SG 3 and SG 4 to detach, making it impossible to measure response signals. Accordingly, the stress measured by strain gauges SG 1 and SG 2 is presented in Figure 18. The maximum stresses were found to be 51.26 MPa for SG 1 and 46.33 MPa for SG 2, both of which are lower than the yield strength of SS 400. Therefore, it is concluded that the structural integrity and safety of the hydrogen storage vessel specimen are satisfactory under seismic loads within the design criteria.

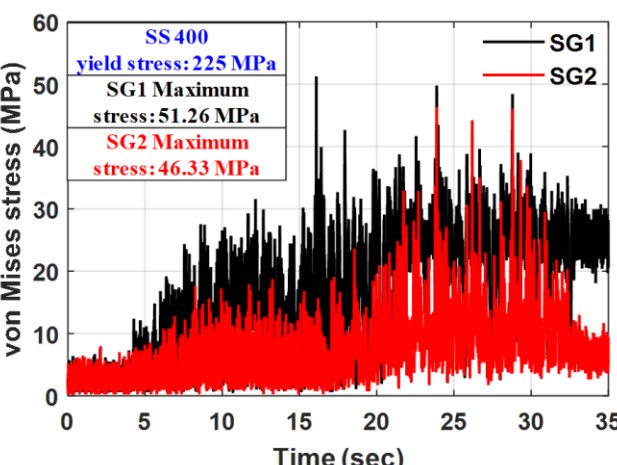

**Figure 18.** The Von Mises stress response results of the supporting column.

## 6. Conclusions

This study aimed to investigate the dynamic behavior and structural safety of a commonly used vertical hydrogen storage vessel under seismic conditions through shaking table testing. To achieve this, an actual site survey was conducted to select the target structure, and the design of the test specimen for the shaking table test was validated through analytical methods. Subsequently, the specimen was fabricated, and a seismic simulation test was conducted on the shaking table. The following conclusions were drawn based on the results of the study:

- Based on the response acceleration of the target structure to the artificial earthquake generated according to the seismic design criteria and analyzed on a time-history basis, it was observed that the seismic motion was amplified by a factor of approximately 10 compared to the original load magnitude. Consequently, it is inferred that the seismic motion occurring during an earthquake could cause significant damage to both the internal and external aspects of the structure, depending on the structure's form and the composition of materials.

- In the analytical verification, the response acceleration tended to decrease as it was transmitted to the point where the weight center of the target structure was located. However, in the actual shaking table test, there was an opposite trend in the acceleration response. This phenomenon is attributed to the slushing effect caused by the seismic motion as the weight of hydrogen is replaced with water. It is considered that additional research on materials that can replace the weight of hydrogen will be necessary in the future.

- The test specimen, a vertical hydrogen storage vessel, was interpreted and experimentally verified to have no structural integrity issues with the main body and anchor in the event of an earthquake. However, the current installation method on protrusion concrete for hydrogen utilization poses a structural risk as a vulnerability when external forces such as earthquakes are applied.

- The failure mode of the test specimen during the shaking table test was similar to the analytical verification results where concrete along the boundary of the support and foundation was damaged due to concentrated settlement stress. In both actual damage cases and analytical verification results under external and seismic loads, concrete fractured along the boundary of the support and foundation, and the concrete at the foundation was completely separated. Therefore, it is recommended to explore alternative approaches considering on-site installation conditions, such as by producing the protrusion concrete element with reinforced concrete or adjusting the height of the foundation concrete to directly install the structure onto the foundation.

**Author Contributions:** Conceptualization, W.J.; Software, S.L.; Validation, S.L.; Investigation, B.J.; Writing—original draft, S.L.; Writing—review & editing, W.J. All authors have read and agreed to the published version of the manuscript.

**Funding:** This research was supported by the "Regional Innovation Strategy (RIS) and Basic Science Research Program" through the National Research Foundation of Korea (NRF) funded by the Ministry of Education (MOE) (2022RIS-005 and 2021R1A6A1A03044326).

**Institutional Review Board Statement:** Not applicable.

**Informed Consent Statement:** Not applicable.

**Data Availability Statement:** Data are contained within the article.

**Conflicts of Interest:** The authors declare no conflict of interest.

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
