# Peer review of "Seismic Performance Evaluation and Analysis of Vertical Hydrogen Storage Vessels Based on Shaking Table Testing"

_applsci, doi:10.3390/app132413190_

Round 1

Reviewer 1 Report

Comments and Suggestions for Authors

The title of this article is Seismic Performance Evaluation and Analysis of Vertical Hydrogen Storage Vessels Based on Shaking Table Testing. This article combines simulation and experimentation to evaluate and analyze the seismic performance of the Vertical Hydrogen Storage Vessel. Before accepting for publication, authors should elaborate on the following content.

1. The structure of the article is clear, including introduction, literature review, methods, results, and discussion. However, in the Methods and Results section, more details and explanations can be provided to help readers better understand the research process and results.

2. The article provides an overview of relevant research in the literature review section, but requires more details and background knowledge to support the importance and innovation of the research. It is recommended that the author provide more relevant research and theoretical foundations in the literature review, so that readers can better understand the background and significance of the research.

3. The author provided some descriptions of the experiment and analysis methods, but more details are needed to ensure the reproducibility of the experiment. Suggest the author to provide more experimental details, including experimental equipment, parameter settings, and data collection methods, so that other researchers can replicate the experiment and verify the results.

4. It is recommended to describe the element size and number of the finite element model.

5. The unit of size is not indicated in Figure 4, and the font in Figure 9 and Figure 13 is not clear. It is recommended to label it clearly.

Author Response

Thank you for your valuable comments to our paper. Please refer to the enclosed attachment for a response to the review.

Reviewer 2 Report

Comments and Suggestions for Authors

The article is interesting, concerning issues related to the safety of a vertical hydrogen storage tank under seismic impact conditions. The authors conducted sclerometric tests of concrete (using Schmidt hammer) to which this type of tanks are attached in-situ. Then, they carried out numerical simulation analyzes of such a tank under the influence of a seismic load and verified the results of these calculations with experimental tests of such a tank on a shaking table. The issue is very interesting, and the results obtained are helpful in designing such structures in seismic areas.

General comments: In general, the introduction is too extensive and multi-threaded. You can successfully remove Fig. 1 and its description without harming the article. This adds nothing to the substantive scope of the article. Moreover, throughout the article, items from the reference list are inadmissibly cited (or not cited). Various citation systems are used, sometimes even mixed. This is not acceptable.

Detailed remakes:

1. Page 3. Lines #91-94. Information regarding previously conducted studies on shaking tables in other centers is important from the point of view of the content of the reviewed article. This should be presented in more detail.

2. Page 3. Line #100. It is first mentioned in the text of ICC AC 156 and is not given a position in the references ([25]).

3. Page 3. Line #119. You need to decide how references are cited: only by giving only the number (e.g. in this case [8]), or the name of the first or two authors and the year of publication. The use of different citation systems in one article is unacceptable. This note applies to the entire article.

4. Pages 3-4. Lines #122-132. Various documents are referenced without the year of publication or number in the references.

5. Page 4. Lines #156-158. It is difficult to formulate a broader summary if only a few items relating to previous research were discussed. Presented sentence is obvious and is not a summary of the literature review on the subject. They need to be reworded.

6. Page 5. Fig.4. The description of this drawing is too brief. The vessel-mounting diagram in case (b) is not very legible. The description should be improved.

7. Page 6. Table 1. How was the relationship between the rebound number and the compressive strength of concrete calibrated? Was the standard relationship used for a given type of Schmidt hammer or a different one? What type of Schmidt hammer was used (e.g. type "N")? The last line of the table shows the average values. In addition, what were the values of standard deviation and CoV?

8. Page 7. Line #222. What is Table 2.2.3.3?

9. Page 7. Table 2. The ASTM A36 standard has no reference number.

10. Page 8. Line #235. The title of the chapter should be "Numerical Verification for Design"

11. Page 9. Lines #268-271. On what basis was, the friction coefficient adopted. A reference to [17] without comment is insufficient.

12. Page 9. Line #280. “Lee and Fevans in 1998 [18]” – an example of confusion between two reference citation systems. There are also such confusions later in the text.

13. Page 9. Line #291. On what basis was, "the elongation ratio is 0.3" adopted.

14. Page 9. Lines #292-293. This is, after all, the e-p-d model developed by Lubliner & Onante.

15. Page 9. Lines #408-424. What parameters did the concrete have in the shaking table test? There is no information in the article, but it may determine the form of destruction (see Fig. 17).

16. Page 14. Line #445. The IEEE 344 standard has no reference number.

17. Page 15. Line #457. What does “fastening force of 1400 kgf·cm” mean? The notation of strength is incomprehensible.

18. Page 19. Line #570. There were no analytical calculations, only numerical ones, and this is a big difference.

Final remarks: The conclusions are generally correct, but they only concern one type of reservoir analyzed. Why, then, did non-destructive testing of concrete for three types of tanks be presented in Chapter 3? It was enough to describe only the Samcheok type vessel, because that was the only one that was calculated and tested on the shaking table. This chapter should be completely rewritten and unnecessary information removed, because it is not substantively related to the numerical and experimental part of the article.

Author Response

(The authors gave the same response as above.)

Reviewer 3 Report

Comments and Suggestions for Authors

Please see the file attached

Comments on the Quality of English Language

Minor editing

Author Response

(The authors gave the same response as above.)

Round 2

Reviewer 1 Report

Comments and Suggestions for Authors

Accept

Author Response

Thank you for accepting and reviewing our paper.

Reviewer 2 Report

Comments and Suggestions for Authors

The reviewer would like to thank you for the explanations sent and the changes as well as improvements introduced to the text of the article. Unfortunately, the authors did not correct the citation of the publication in the article carefully. There are still numerous irregularities in the text:

1. Page 3. Line #129. There is still "Zhou (2019)" while there is no such item in the references at all. There are two multi-authored publications in the references, in which Zhou is a co-author (it is not known whether it is the same person). These are items [7] from 2015 and [9] from 2018.

2. Page 4. Line #138. The situation is similar with the citation of "Nam (2012)". The references include [5] Nam et al. but from 2010. This is probably not this article.

3. Page 4. Line #148. "Maekawa (2010)" is cited, which does not appear anywhere in the references at all.

4. Page 4. Line #157. The work cited is "Silvestri (2022)". There is no such item in the references list. There are only two items where Silvestri is a co-author. This is probably reference [8] from 2022, but the correct citation should then be "Furinghetti et al. [8] …...” If the name of the author(s) is given, it is always the first author. Unless, however it is a different article?

5. Page 4. Lines #165-166. Silvestri is cited without a reference number (see note above). The European Standard EN 1998-4 has no number because it is not included in the references.

Author Response

(The authors gave the same response as above.)
